# Targeting Antigens for Universal Influenza Vaccine Development

**DOI:** 10.3390/v13060973

**Published:** 2021-05-24

**Authors:** Quyen-Thi Nguyen, Young-Ki Choi

**Affiliations:** 1College of Medicine and Medical Research Institute, Chungbuk National University, Cheongju 28644, Korea; quyenbio@gmail.com; 2Zoonotic Infectious Diseases Research Center, Chungbuk National University, Cheongju 28644, Korea

**Keywords:** influenza, universal vaccine, antigen, immune response

## Abstract

Traditional influenza vaccines generate strain-specific antibodies which cannot provide protection against divergent influenza virus strains. Further, due to frequent antigenic shifts and drift of influenza viruses, annual reformulation and revaccination are required in order to match circulating strains. Thus, the development of a universal influenza vaccine (UIV) is critical for long-term protection against all seasonal influenza virus strains, as well as to provide protection against a potential pandemic virus. One of the most important strategies in the development of UIVs is the selection of optimal targeting antigens to generate broadly cross-reactive neutralizing antibodies or cross-reactive T cell responses against divergent influenza virus strains. However, each type of target antigen for UIVs has advantages and limitations for the generation of sufficient immune responses against divergent influenza viruses. Herein, we review current strategies and perspectives regarding the use of antigens, including hemagglutinin, neuraminidase, matrix proteins, and internal proteins, for universal influenza vaccine development.

## 1. Introduction

Influenza viruses are enveloped RNA viruses belonging to the *Orthomyxoviridae* family. There are four genera of influenza viruses, A, B, C, and D, divided based on antigenic differences [1]. Among influenza types, human influenza A and B viruses both cause seasonal flu, while influenza A viruses (IAVs) are the only viruses to have caused pandemics. The IAVs are divided into two phylogenic groups, 1 and 2. Group 1 viruses comprise H1, H2, H5, H6, H8, H9, H11, H12, H13, H16, H17, and H18, while group 2 viruses contain H3, H4, H7, H10, H14, and H15. Influenza B viruses are categorized into two lineages, B/Yamagata-like and B/Victoria [2]. Influenza C viruses are less prevalent and reportedly cause only mild disease in humans, whereas influenza D viruses are not known to infect humans [3], but infect cattle, swine, ruminants, horses, and camels [1,4].

Vaccination is considered the most effective way to control influenza infections, although existing influenza vaccines have some limitations. Currently, there are three main types of seasonal influenza vaccines, live attenuated, inactivated, and recombinant. Live attenuated influenza vaccines (LAIVs) have been considered as alternatives to traditional inactivated influenza vaccines due to their ability to mimic a natural infection and induce humoral, cellular and mucosal immune responses [5]. The LAIV was first used in Russia over 40 years ago, was licenced in the United States in 2003 and in Europe in 2012 [5], and has been demonstrated to induce protection against antigenic variants of influenza A viruses [6,7,8,9]. The influenza virus strains targeted by seasonal vaccines are selected based on global surveillance coordinated by the World Health Organization; thus, vaccine effectiveness is greatly variable depending on the match between the vaccine strains and circulating viruses. Due to the rapid antigenic shift and drift of the influenza hemagglutinin (HA) and neuraminidase (NA) glycoproteins, reformulation and readministration of the vaccine is required annually. Although seasonal influenza vaccines are updated yearly to match circulating strains, vaccine effectiveness was only 10 to 60% for flu seasons from 2004–2020 [10] (Figure 1), resulting in 250,000 to 500,000 deaths every year as estimated by the World Health Organization [11]. Based on the preliminary end-of-season estimates, the influenza vaccine effectiveness (VE) estimate for the 2019–2020 flu season was only 39% [10]. In addition to seasonal epidemics, influenza pandemics are unpredictable and significant global threats. The four biggest pandemics of the 20th and 21st centuries, the 1918 Spanish (H1N1 virus), 1957 Asian (H2N2 virus), 1968 Hong Kong (H3N2 virus), and 2009 swine (pandemic H1N1 virus) flu, caused global health crises with significant mortality and morbidity and enormous economic burdens. The recent influenza pandemic in 2009, explicitly demonstrated how the influenza vaccine was insufficient for controlling a potential pandemic as well as seasonal epidemics. Thus, there is a need to improve the immunogenicity and efficacy of current influenza vaccines. In addition, COVID-19, caused by the SARS-CoV-2 virus and first appearing in late December 2019, still threatens health globally with increasing numbers of infected patients and deaths. Further, coinfection with influenza more than doubles the risk of death in COVID-19 patients [12]. Thus, an effective influenza vaccine is crucial to limiting severe outcomes of COVID-19 when coinfection occurs.

Numerous efforts have been made to develop universal influenza vaccines (UIVs) that are able to provide at least 75% effectiveness against symptomatic disease caused by group 1 and 2 influenza viruses with durable protection lasting at least 12 months in all populations [13]; however, currently, there is no commercial UIV available. Numerous approaches have been explored, including the targeting of conserved antigens such as HA, NA, matrix, and internal proteins, together with the application of multiple vaccine platforms, including recombinant antigen/protein-based vaccines [14], reassortant/recombinant influenza virus-based vaccines [15], virus-vectored vaccines [16], virus-like particle (VLP) vaccines [17], nanoparticle-based vaccines [18,19], DNA/RNA-based vaccines [20,21], and multiple platform vaccines [22,23]. Careful selection of target antigens for UIV development is imperative due to the need to generate broad immune responses against divergent seasonal and potential pandemic influenza virus strains. In this review, we summarize advantages and limitations of current antigen targeting strategies for UIV development.

## 2. HA-Based UIVs

Influenza HA molecules consist of two distinct domains, a globular head, composed of part of HA1, and a stalk structure, composed of portions of HA1 and all of HA2. Traditional IAV vaccines depend on the generation of neutralizing antibodies (Abs) specific for the variable HA head domain, and thus the vaccines are strain-specific. Current strategies target less variable HA regions, including the stalk domain or/and conserved epitopes within the head domain. These approaches include the use of headless HA, chimeric HA, mosaic HA, computationally-optimized broadly reactive antigens (COBRA), and “breathing” HA.

### 2.1. Headless HA

The discovery of broadly neutralizing Abs against the conserved stalk domain has spurred the development of stalk domain-based UIVs. The first attempt to produce immunosubdominant headless HA was made by Graves et al. in 1983 by the chemical removal of the immunodominant HA global head domain [24]. However, this treatment results in conformation changes in the HA2 stalk domain and removal of conformational epitopes in the HA1 stalk domain, which are needed for the binding of broadly reactive anti-stalk Abs [24]. Other efforts, including genetic treatment and expression of headless HA in *E. coli* or insect cells, have also been made; however, the conformation of the headless HA is likely far from optimal despite the fact that it provided homologous, heterologous, and heterosubtypic protection after virus challenges [25,26,27]. In 2015, Impagliazzo et al. [28] and Yassine et al. [29] independently generated stable and correctly folded headless HA by introducing a stabilizing element in the membrane-proximal end of the molecule. The headless HA induced stalk-specific Abs and protected animals against heterosubtypic challenge with H5N1 viruses via Fc-dependent mechanisms, such as antigen-dependent cellular cytotoxicity (ADCC), without robust neutralizing Ab titers [28,29]. These findings not only show the potential importance of headless HA in UIV development, but an approach to overcome conformation changes in this molecule. However, constructs based on the H1 HA stalk only provided protection against group 1 IAVs. In 2016, Poon’s group first reported that a group 1 headless HA mini-stem conferred protection against both group 1 (H1 and H5) and group 2 (H3) influenza viruses [30]. Moreover, a recent study by Deng et al. showed the use of a headless HA formulated with a tetrameric M2e into protein nanoparticles induced long-lasting immune responses and protected mice from challenges with divergent IAVs of the same group or of both groups [19]. While continuous efforts have been made to develop UIVs based on headless HA, there are some concerns about the induction of antibody-dependent enhancement (ADE) by HA stem-specific Abs [31,32]. Although the above studies warrant further exploration of headless HA-based UIVs, it is a prerequisite to carefully evaluate whether headless HA-induced Abs could enhance ADE, thereby increasing disease severity.

### 2.2. Chimeric HA

Aside from headless HA, another strategy to refocus the immune response from the HA head towards the HA stalk is to use a chimeric HA (cHA). This approach involves the induction of anti-stalk Abs through sequential vaccination with cHAs sharing the same stalk domain, but different head domains (Figure 2). The cHA consists of the HA stalk domain from current seasonal IAVs, such as H1, H3, or influenza B, and an HA head domain from an exotic avian virus [33]. The first vaccination with a cHA induces Abs specific to both the head and stalk domains. Boosting with another cHA that contains the same stalk domain as the first vaccination and a different head domain leads to the induction of a primary response to the new head domain and a recall response to the fixed stalk domain. The anti-stalk response can be further boosted by additional immunization with yet another cHA consisting of the fixed HA stalk domain and a different HA head domain [34]. Vaccination with cHA in various vaccine formulations, such as recombinant protein, viral vector or nucleotide-based vaccines, has been shown to induce highly stalk-reactive Abs and to consequently protect mice against challenge with divergent IAVs [35,36,37,38,39,40]. Further, sequential administration of the cHA vaccines (cH8/1 and cH5/1) was demonstrated to be locally and systemically well-tolerated in rabbits [41]. Since this strategy induces both anti-head and anti-stalk-reactive Abs, the protection conferred by cHA is mediated by both neutralizing and non-neutralizing Abs (such as ADCC activity) [35,36,37,38,39,40]. Treatment of ferrets with cHA-based vaccines conferred protection against pandemic H1N1 virus infection [42]. Sequential administration of cHA-based vaccines with live-attenuated vaccines also induced HA stalk-specific specific humoral responses, conferred protective heterosubtypic immunity against different IAVs in the ferret model [43,44], and induced long-lived immune responses against pandemic H1N1 virus infection [43]. Recently, phase I clinical trials (NCT03300050) using cHA-based UIV revealed that the vaccine was safe for human use (Table 1). Healthy adults immunized with the vaccine exhibited remarkably high anti-stalk Ab titers and long-lasting immune responses [43,45].

Although cHA-based vaccines could induce Abs against anti-stalk HA and provide protection against both homologous and heterologous viral challenges, the protective efficacy of this vaccine might still be limited depending on the chimeric antigen subtypes. Further, multiple-dose vaccinations of different chimeric vaccines are required to induce vaccine efficacy with a broad spectrum. Therefore, planned antigen selection and simplified vaccine strategies are needed for further study.

### 2.3. Mosaic HA

Mosaic HAs (mHAs) were designed with the idea of eliciting antibodies not only against the conserved stalk domain but also against epitopes in the head domain outside of the major antigenic sites. Given that some monoclonal Abs showed cross-reactivity with the HA head domain of different influenza strains [46,47,48], mHA was developed to maximize cross-reactive responses and minimize genetic differences. The mHA can induce immune responses toward both the head and stalk domain of HA (Figure 3). This differs from cHA in which the entire HA head region was replaced. Specifically, mHA is constructed by replacement of only major antigenic sites of the head HA with those from exotic avian HA subtypes. Thus, the conserved head and stalk domains of mHA are derived from H1, H3, or influenza B virus HA. Priming with mHA induces high amounts of Abs against antigenic sites of the HA head domain and low amounts of Abs specific for conserved epitopes in the HA stalk domain. However, boosting with another mHA (differing in antigenic sites) induces Abs against both antigenic sites and conserved epitopes in the HA head and stalk domains [34]. In the mouse model, sequential immunization with mosaic influenza B HA was found to induce cross-protection against both homologous and heterologous influenza B virus strains [49]. Vaccination with mosaic H5 HA enhanced antibody and T cell responses, providing protection against H5N1 and H1N1 viruses in mice [50,51]. Mice vaccinated with H1 mosaic HA or H3 mosaic HA also exhibited broadly protective immune responses against divergent influenza virus strains [52,53]. Kingstad-Bakkle and colleagues showed that a mosaic H5 vaccine antigen delivered via polyanhydride nanoparticles or modified virus Ankara induced humoral and cellular immune responses in both specific-pathogen-free (SPF) and commercial chicks [54]. A study by Florek et al. revealed that immunization with mosaic H5 HA increased virus clearance and elicited cross-reactive antibodies capable of mediating ADCC activity in rhesus macaques [55]. Recently, a mosaic receptor-binding domain (RBD) nanoparticle displaying multiple copies of eight different HA head domains from H1 HA antigens was developed at the Vaccine Research Center of the National Institute of Allergy and Infectious Diseases (United States). Vaccination with these mosaic RBD nanoparticles elicited cross-reactive B cell responses in mice [56].

Although mHA-based vaccines could provide broader cross-reactive protection against both homologous and heterologous viral challenges, similar to cHA-based vaccines, the protective efficacy of this vaccine might still be limited depending on multi-dose vaccination and selected antigen subtypes. Therefore, planned antigen selection and simplified vaccine strategies are needed for further study.

### 2.4. COBRA HA

The COBRA strategy applies a computational method to generate a consensus sequence for all strains from a certain HA subtype [57]. The main purpose of this vaccine strategy is to generate a broad anti-HA-head response with enhanced HI [hemagglutination inhibition] activity. The COBRA HA approach has been demonstrated to be an effective strategy for the development of influenza vaccines against panels of H1N1, H5N1, and H3N2 viruses by eliciting broadly reactive Abs in mice, ferrets, and non-human primates [57,58,59,60]. Carter et al. reported that COBRA H1 HA antigens could generate anti-head Abs against 17 human seasonal and pandemic H1N1 viruses isolated since 1918 [58]. A recent study revealed that the use of COBRA HA antigens designed using both human and swine H1 HA sequences protected mice against viruses of both swine and human origin [61]. In addition, animal studies by Giles BM et al. have shown that COBRA H5 HA antigens elicit broadly reactive Abs with HI activity against 25 highly pathogenic avian influenza H5N1 virus strains [57,59]. COBRA H3 HA antigens have been demonstrated to induce HI antibodies to protect mice against a panel of H3N2 influenza virus cocirculating variants [60]. More recently, Ross et al. developed a COBRA H3 HA vaccine by generating a consensus sequence from 22,144 human A(H3N2) viruses collected from 2002 to 2015 [62]. This vaccine candidate showed an increase in HI activity compared to wild-type HA vaccine. COBRA H2 HA antigens were also developed using human and avian H2 HA sequences, and the use of this vaccine generated broadly cross-reactive Abs against both historical human H2N2, avian, and mammalian H2 viruses recently isolated in mice [63].

Overall, the COBRA HA approach is one of the promising strategies to develop UIVs by generating broadly reactive anti-head HA Abs to protect against a broad range of influenza viruses. However, sometimes the generated consensus sequences do not generate fully functional HA proteins, and thus it needs to use alternative vaccine platforms, such as VLP. Currently, COBRA HA vaccines are in the late preclinical stage of development, and clinical trial studies are needed to investigate their vaccine effects in humans.

### 2.5. “Breathing” HA

More recent advances in HA-based vaccines utilize targeting of hidden, conserved epitopes in the HA domain, called the “breathing” HA strategy [64]. Studies by Bangaru et al. [65] and Watanabe et al. [66] have shown that Abs directed against concealed, conserved epitopes in the HA head domain interface could provide protection against broad-spectrum IAV subtypes. For instance, a human monoclonal antibody, FluA-20, recognizes epitopes positioned in the 220 loop and the adjacent 90 loop, which is usually buried in the native HA trimer. This antibody protects mice from sublethal and lethal challenges with human H1N1, H5N1, H3N2, and H7N9 viruses [65]. The collective breadth of Abs (S5V2-19, H2214, S1V2-58, S8V2-17, and S8V2-37) isolated from memory B cells of inactivated seasonal influenza vaccine-immunized donors target novel sites at the interface of the HA domain and provide protection against both group 1 (H1, H2, H5, and H9) and 2 (H3, H7, and H14) viruses in mice [66]. However, a method to expose the hidden epitopes will require further exploration. Glycosylation of the HA domain is reportedly involved in the antigenic drift of IAVs, allowing escape from the host immune response [67,68,69]. Interestingly, the glycosylation site of the HA head domain can be modified to expose the hidden conserved epitopes in order to direct immune responses to the conserved HA domain. Previous studies have reported that mice vaccinated with the hyperglycosylated HA head domain exhibit enhanced, conserved stalk domain-directed antibodies and have better protection against lethal virus challenge than those vaccinated with wild-type HA [70]. Studies by Lin et al. have shown that vaccination of hyperglycosylated HA in various formulations, such as VLPs, recombinant protein, or adenoviral vector, elicits broadly neutralizing Abs against avian influenza H5N1 viruses [71,72]. Another study revealed that vaccination of glycan-shielded HA antigens stably expressed by CHO cell clones increased potential neutralizing Abs against heterologous H5N1 viruses [73]. Hyperglycosylation of the HA domain does not dampen overall humoral responses, but rather changes patterns of immunodominance and elicits broadly protective Abs, as reported by Bajic et al. [74]. Collectively, these studies indicate that hyperglycosylation could be a promising strategy to find hidden epitopes in the HA head and stalk domains, facilitating the development of novel UIVs in the future. Currently, glycosylation in influenza vaccine design is still under debate. Studies showed that glycosylation was important for virus infectivity and host immune responses. Wu et al. reported that glycosylation of HA at N-142 promoted H1N1 influenza virus infectivity and the glycosite 142 was important for the virus in evading immune responses in humans [75]. The addition of N-link glycosylation modification to the head HA region also decreased Ab titers in HI assay and plaque-reduction neutralization test [76]. Thus, it is necessary to determine the exact glycosites needed for influenza vaccine development. Various studies should also be performed to have a better understanding of the influence of glycosylation on vaccine efficacy.

## 3. NA-Based UIVs

The second major glycoprotein, NA, plays an important role in facilitating influenza virion release from infected cells through its sialidase and neuraminidase activities. The enzymatic activity of NA is a crucial target for antiviral drugs, such as Oseltamivir and Zanamivir [77]. Similar to HA, NA also undergoes antigenic drift; however, at a slower rate [78]. Moreover, studies in mice and humans have shown that anti-NA Abs wane at slower rates compared to anti-HA Abs [79,80].

Recently, the influenza virus NA is emerging as a target of broadly protective Abs recognizing its active site [81,82]. Stadlbauer et al. described human mAbs targeting NA that could neutralize the influenza A virus, inhibit neuraminidase activity, and provide broad in vivo protection against challenge with group 1 (a human N1 and avian N1, N4, N5, and N8), group 2 (a human N2, a swine N3, avian N2, N6, N7, and N9), and an influenza B (B/Victoria/2/87 lineage) virus in mice [81]. More recently, Madsen et al. showed human mAbs could also neutralize different influenza B viruses (IBVs), inhibit the NA neuraminidase activity, induce ADCC activity, and protect mice against IBV challenge [82]. With these novel findings, NA might be an attractive target for UIVs.

In inactivated vaccines, NA is varied in quality and quantity [83]. Further, these vaccines induce anti-NA responses with approximately 30% seroconversion [84,85,86]. A previous study reported that NA-specific Abs could inhibit influenza virus replication by interfering with viral egress [87]. In addition, mAbs targeting conserved NA epitopes could also induce ADCC activity contributing to the protection against influenza B virus in mice [88]. Doyle et al. reported that a mAb targeting a conserved NA sequence of IAV provided heterosubtypic protection in mice challenged with lethal doses of H1N1 and H3N2 viruses [89]. Recently, guinea pigs vaccinated with recombinant influenza B NA showed reduced virus titers, a high level of anti-NA Abs, and a decrease in virus transmission [90]. Further, it has been demonstrated that some H3N2 viruses use NA instead of HA for cell attachment [91,92,93], suggesting that NA Abs might help to block virus attachment. Further studies to address this hypothesis are needed.

In spite of NA’s benefits, it is frequently ignored as a target in vaccine development and has not been approved as an effective vaccine antigen due to standardization issues, including an unsuitable method for quantification of NA content and the lack of an easy assay to measure NA Abs [83,94,95]. Of note, NA content in vaccines is likely to be associated with anti-NA Ab response [96], although the presence of immunodominant HA head domains in the vaccines could lead NA to become immunosubdominant [97,98]. Increasing the amount of NA in vaccines accompanied by improving methods for NA quantitation could help to improve anti-NA immunity. Furthermore, neuraminidase activity in stored vaccine lots decreases over time depending on the strains used [99,100]; thus, specific storage conditions are needed to maintain its activity.

To our knowledge, there is no clinical trial of NA-based vaccines carried out so far; thus, the development of the vaccine becomes more challenging and requires more effort to make them available. Clinical trials and observational studies of NA-based vaccines should be performed to evaluate their vaccine effects in humans.

## 4. M2e-Based UIVs

Matrix protein 2 (M2) is a transmembrane protein which can be divided into three parts: the extracellular N-terminal domain (M2e), the transmembrane domain, and the intracellular C-terminal domain [101]. The M2 protein is essential for influenza A virus fitness and is crucial for the uncoating process occurring in endosomes after viral entry [102]. Since M2e is highly conserved, it is a potential target for UIVs. However, compared to HA and NA, M2e is smaller and less immunogenic, resulting in low immune responses when given alone. To resolve this issue, researchers have tried to increase the immunogenicity of M2e by incorporating carrier proteins, using nanoparticle formulations, presenting on VLP, including other influenza proteins, or adding adjuvants [103]. It is believed that anti-M2e Abs cannot neutralize viruses; however, anti-M2e Abs can bind to M2e protein expressed in infected cells and reduce virus replication by inhibiting viral budding. Other mechanisms conferring protection include anti-M2e Ab-mediated cell killing to eliminate influenza virus-infected cells through complement-dependent cytolysis, ADCC, and/or antibody-dependent cellular phagocytosis [104]. M2e-specific T cells can also mediate protection against influenza infection [104].

In animal models, multiple formulations of M2e-based vaccines showed cross-protection against different influenza virus strains. In mice, vaccination with M2e displayed on recombinant, *Escherichia coli*-derived outer membrane vesicles can induce anti-M2e immunity to control influenza A virus replication as well as provide 100% protection against lethal challenges with H1N1 and H3N2 [105]. A VLP vaccine containing a tandem repeat of the M2e sequence from human, swine, and avian IAVs induced recruitment of cells (macrophages, monocytes, neutrophils, and CD11b^+^ dendritic cells) and production of inflammatory cytokines and chemokines at the site of infection [106], conferring protection against H1N1, H3N2, and H5N1 IAV subtypes [107]. Studies showed that a fusion protein containing tandem repeats of the M2e sequence fused with the N-terminus of an adjuvant fragment of *Mycobacterial* HSP70 enhanced both humoral and cellular immunity, reduced virus shedding, provided protection against H1N1, H3N2, and H9N2 virus challenges in mice and H9N2 challenge in chickens [108,109]. A live bacterial vaccine comprised of M2e expressed on the surface of *Lactococcus lactis* decreased viral burden and prolonged survival of vaccinated chickens after the H5N2 virus challenge [110]. Moreover, chickens vaccinated with different M2e epitopes exhibited protection against H5N1 [111]. Although few studies of M2e vaccine efficacy in ferrets have been reported, immunization with M2e conjugated with carriers increased anti-M2e IgG Abs and reduced viral shedding in ferret lungs after H1N1 viral challenge [112]. In addition, vaccination with M2e displayed on *Escherichia coli*-derived outer membrane vesicles was recently shown to reduce lung virus titers after pandemic H1N1 virus challenge in ferrets [105].

In humans, a phase I clinical trial of a tandem repeat of M2e fused to the hepatitis B core protein (NCT00819013) showed the induction of anti-M2e Abs, although the Ab titers decreased over time [113]. A recombinant M2e–flagellin influenza vaccine (STF2.4xM2e or VAX102) was demonstrated to be safe and induced high anti-M2e Ab levels in healthy adults (18–49 years old) [114]; however, this vaccine candidate is no longer under development. Another study showed that after 24 h of H3N2 virus challenge, healthy volunteers intravenously administered mAb against M2e (TCN-032) (NCT01719874) had reduced symptom scores and virus replication compared to those treated with placebo [115]. Besides studies using M2e as a vaccine antigen, Flugen Inc. (United State) developed a live attenuated, single replication virus vaccine deficient for the M2 gene (Table 1). The safety and immunogenicity of this vaccine were investigated in healthy adults (NCT03999554, NCT02822105, NCT03553940, clinicaltrials.gov) and are currently being investigated in adults ages 50–85 years old (NCT04785794, clinicaltrials.gov). Although M2e-based vaccines showed some promising results, none have achieved commercialization. Therefore, expanded clinical studies of the efficacy of M2e-based vaccines are still needed to provide evidence that these vaccines can protect humans from different influenza viruses.

M2e-based vaccines have some limitations. Anti-M2e Abs can protect against influenza A viruses but not influenza B viruses because the M2 protein in IBV is structurally different [101]. Further, due to low immunogenicity, M2e might not be a standalone, universal vaccine but may need to be given with carriers or other vaccines such as seasonal vaccines. In addition, it has been shown that M2e fused to carrier proteins can induce immune responses specific to the carrier protein, which may cause unexpected side effects such as local and systemic adverse effects, when given in humans at high doses [114]. Furthermore, clinical trials in humans showed a decrease in Ab titer over time, suggesting that M2e should be used in concert with other influenza A antigens or adjuvants to achieve sustainable immune responses.

## 5. Internal Proteins-Based UIVs

Internal proteins of influenza viruses, such as matrix protein 1 (M1) and nucleoprotein (NP), are highly conserved. In contrast to HA, NA, and M2e, which are expressed on the surface of infected cells allowing accessibility of specific Abs, M1, and NP are produced inside infected cells, processed, and presented by major histocompatibility complex molecules for T cell recognition [116]. Thus, targeting internal proteins is also a promising strategy to improve current influenza vaccines by enhancing cross-reactive T cell responses.

In mice, intranasal immunization of *E. coli*-expressed M1 protein with chitosan adjuvant provided 70% and 30% protection against heterologous H1N1 and H5N1 viruses, respectively [117]. Liu et al. showed that mice primed with a DNA-based vaccine containing M1 and boosted with a recombinant M1 protein from avian H9N2 induced humoral and cellular immune responses and provided complete protection against homologous virus infection and partial protection against heterosubtypic H1N1 virus infection [118]. Furthermore, NP-based DNA vaccines were demonstrated to provide protection against both homologous and heterologous influenza viruses in animal models decades ago [119]. Recently, NP-based mRNA vaccines have become an attractive approach. Vaccination with an mRNA vaccine encoding NP induced cross-strain immunity against influenza virus in mice [120]. Moreover, the recombinant protein TAT-NP which generated by incorporation of a protein transduction domain, TAT, from human immunodeficiency virus-1 (HIV-1) into NP, enhanced cellular immune responses and increased protective efficacy against homologous PR8, heterosubtypic H9N2, and H3N2 IAV in mice [121]. Campo et al. demonstrated that mice immunized with a recombinant NP vaccine, OVX836, exhibited T cell responses and protective efficacy against H1N1 and H3N2 IAVs [122]. Aged mice vaccinated with NP adjuvanted with a second-generation lipid adjuvant in stable emulsion (SLA-SE) showed enhanced viral clearance and survival rates after lethal challenge with PR8 virus [123]. Vaccination of a fusion of NP+M1+ heat shock protein 60 mixed with an oil-in-water adjuvant induced robust humoral, mucosal, and cellular immune responses, inhibited viral replication in lungs, and completely protected mice from challenge with H7N9 [124]. Another study showed that the modified vaccinia virus Ankara (MVA) vector encoding NP and M1 (MVA−NP+M1) acted as an adjuvant to enhance Ab and T cell responses in mice, chicken, and pigs [125].

The MVA−NP+M1 vaccine candidate has been tested in several clinical trials since 2011 [126], (Table 1). MVA−NP+M1-vaccinated volunteers exhibited reduced influenza symptoms and length of virus shedding [127]. Another clinical trial showed that MVA−NP+M1 is safe in humans and induces antigen-specific T cell responses [128,129]. The safety and immunogenicity of the vaccine were also confirmed in adults over 50 years of age, as shown by its ability to boost memory T cells and induce multifunctional cytokines [130]. Further, the combination of this vaccine with the seasonal influenza vaccine was demonstrated to increase influenza virus-specific Ab responses and memory T cells in adults aged 50–85 years [131]. The MVA-NP+M1 vaccine is in a phase IIb clinical trial (NCT03880474) to investigate its efficacy in adults aged 18 and older [132]. However, this trial is being stopped for futility (updated on 26 April 2021, clinicaltrials.gov). Vaccination with a novel recombinant simian adenovirus, ChAdOx1 NP+M1, was also found to induce increased T-cell immunogenicity [133], (Table 1). The vaccine candidate OVX836 based on NP oligomer was also tested in phase I clinical trials (NCT04192500 and NCT03594890, clinicaltrials.com, accessed on 28 March 2021), though results have not yet been reported (Table 1).

Although internal protein-based influenza vaccines showed promising results concerning the induction of T cell immune responses and cross-protection, however, internal protein-based vaccines weakly induce Ab response, which is needed to block influenza virus infection. In addition, a major challenge in developing T cell-based vaccines based on internal protein is the diversity of HLA haplotypes responsible for antigen peptide binding and T cell presentation. Thus, to produce effective internal protein-based vaccines that provide sufficient coverage to all individuals with different HLA diversity, there is a need to further evaluate the response in the context of ethnicity [134].

## 6. Multiple Proteins/Peptides-Based UIVs

Given the potential of different protein-based vaccines, a vaccine comprising multiple proteins or epitope peptides is of interest for the potential to boost both broadly cross-reactive antibody and T cell responses. This vaccine strategy has been studied with many different vaccine platforms and protein combinations. Immunization with a combination of a virus-vectored vaccine expressing NP and M1 (ChAdOx1 NP+M1) and a cHA vaccine induced Ab responses specific for HA, NP, and M1 of divergent group 2 IAV, enhanced T cell responses to NP and M1, and protected mice against H3N2 virus challenge [135]. Park et al. reported a recombinant, live-attenuated H3N2 virus expressing M2 and cHA that protected mice against a broad range of IAVs, including H1N1, H3N2, H5N1, H7N9, and H9N2 [136]. In addition, chimeric subunit vaccines containing conserved HA, M2e, and NP proteins were shown to protect mice from challenges with homologous and heterologous IAVs [137,138]. Moreover, mice immunized with double-layered protein nanoparticles containing M2e and HA were protected from challenge with H1N1, H3N2, H5N1, and H7N9 viruses [19]. Further, an M1-HA2 fusion protein expressed by recombinant *Lactococcus lactis* provided mucosal protection against the H9N2 virus in chickens [139].

Vaccine candidates developed using this strategy have also been tested in many clinical trials. Multimeric-001 (M-001) developed by BiondVax Pharmaceuticals Ltd. (Israel) is a peptide-based vaccine containing B and T cell epitopes from the HA, M1, and NP proteins [140]. In fact, M-001 has already been assessed in seven clinical trials with promising results (Table 1). A phase II trial of M-001 (NCT03450915) showed that standalone M-001 was a primer for an HA-based vaccine, and elevated HI titers in the elderly [141]. In addition, M-001 can be used as a primer to enhance the HI response to a trivalent influenza vaccine [142], as well as a primer or as a standalone vaccine for H5N1 IAV [143]. Of note, M-001 was also evaluated for safety and clinical efficacy in phase III clinical trial as a standalone UIV in adults ≥ 50 years of age [144].

FP-01.1, developed by Immune Targeting Systems Ltd. (United Kingdom), is a synthetic peptide-based vaccine containing six peptides derived from the NP, M, and polymerase basic 1 and 2 proteins of influenza A (Table 1). This vaccine was tested in four phase I clinical trials [145,146,147] and showed good safety and tolerability profiles [148]. However, FP-01.1 is no longer in development.

FLU-v is a synthetic peptide vaccine composed of NP, M1, and M2 proteins developed by PepTcell (SEEK, United Kingdom) (Table 1). FLU-v has been thoroughly investigated in four phase I and II clinical trials. The findings from the phase I study showed that FLU-v could stimulate cell-mediated immunity [149], and reduce symptomatology and virus shedding [150]. Results from clinical phase II studies revealed that FLU-v reduced viral shedding as well as clinical influenza symptoms and induced long-term cell-mediated immunity [151,152]. However, the efficacy of this vaccine still needs to be further explored.

Overall, the multiple proteins/peptides-based vaccine approach is one of the promising strategies to develop UIVs by combinations of multiple proteins or peptides, but there are still limitations, such as the sequence variations of target proteins among the subtypes. Therefore, the sequences of target proteins in designed vaccine candidates should be screened to provide broad protection against different virus strains. The immunodominant regions should also be well selected to obtain the most appropriate epitopes for vaccine construct.

## 7. Influenza Vaccine Adjuvants

One approach to improve influenza vaccine effectiveness is to include adjuvants. Three main adjuvants, aluminum salts (alum), MF59, and AS03, are incorporated in licensed influenza vaccines [153].

Since the 1920s, alum has been used in a wide range of vaccines in the United State and Europe [154,155]. Previous studies demonstrated that alum stimulates Th2-biased responses through mechanisms including the depot effect [156], NLRF3 inflammation activation [156], stimulation and differentiation of CD4^+^ T cells [157], perturbation of dendritic cell membrane [158], and complement activation [159]. Alum is well-known to trigger robust humoral immune responses; however, it only minimally induces ADCC [160,161,162]. In addition, alum only weakly elicits cellular immune responses [154,163].

Licensed in Italy in 1997, MF59 was developed as an adjuvant for influenza vaccination in the elderly [164]. Licensed influenza vaccines, including Focetria, Celtura, and FluAd as well as seasonal influenza vaccines, all include MF59 [153]. Studies have demonstrated that MF59 significantly enhances antigen-specific antibody production and boosts both Th1 and Th2-biased responses [165,166]. However, MF59 can cause pain at the injection site, reactogenicity, and can induce inflammatory arthritis [167].

The AS03 adjuvant has been used in licensed vaccines Pandemrix, Arepanrix, Prepandrix, and Q-Pan H5N1 [153,168]. Previous studies showed that AS03 induces the production of inflammatory cytokines and chemokines in injected muscles and draining lymph nodes and enhances migration of monocytes and dendritic cells into draining lymph nodes. Additionally, inclusion of AS03 results in persistent production of neutralizing antibodies and a high frequency of memory B cells [169]. However, studies reported that AS03-adjuvanted vaccines could cause sleeping disorders and narcolepsy after vaccination [170,171]. Furthermore, several strategies, such as the use of toll-like receptor agonists or different combinations of adjuvants, have been tested to overcome the limitations of the common adjuvants [153,172].

In summary, each adjuvant has unique advantages and disadvantages. Thus, adjuvant selection for influenza vaccines is crucial to boost the immunogenicity of vaccine antigens and increase vaccine efficacy. However, safer and more effective adjuvants are needed to effectively improve both humoral and cellular immune responses specific to vaccine antigens.

## 8. Conclusions

The continuous threat of an influenza pandemic underscores the necessity to develop novel UIVs with broader and greater protective efficacy than prevailing influenza vaccines against seasonal and potential pandemic strains. Nonetheless, despite continuous efforts, there are currently no commercial UIVs available. Although various influenza antigens are being testing for UIV development, different vaccine antigens have unique advantages and disadvantages. The use of HA-based vaccines mainly elicits Ab responses, headless and chimeric HA vaccines focus on responses targeting the conserved HA stem region, while mosaic vaccines target both head and conserved HA regions. In addition, the computationally-optimized broadly reactive antigens (COBRA) strategy generates a consensus sequence which allows elicitation of broadly reactive Abs against different virus strains. Furthermore, the breathing HA approach targeting the concealed HA region also shows promise. Current seasonal influenza vaccines use NA together with HA as antigens. Although NA-based vaccines have shown induction in immune responses, NA is currently ignored as an antigen target in influenza vaccine development due to its limitations, such as standardization issues. Further studies of NA-based influenza vaccines should be performed to demonstrate the important roles of NA as an influenza vaccine antigen. Additionally, advanced technologies should be developed to quantify the absolute amount of NA and measure NA Ab responses accurately and maintain neuraminidase activity of the NA-based vaccines during storage. Although M2e, a well-conserved protein of influenza viruses, is a promising target for UIVs, the M2e protein has low immunogenicity. Therefore, to induce robust immune responses, it should be used with other viral proteins or adjuvants. In addition, the internal protein-based vaccines, that mainly induce T cell responses, facilitating the killing of influenza-infected cells, is one of the promising strategies to develop UIVs by combinations of multiple proteins or peptides; however, these vaccines weakly induce Ab response, hereof the immunodominant regions should be properly selected to obtain the most appropriate epitopes for vaccine construct.

Because of their capacity to enhance the immunogenicity of vaccines, adjuvants have been widely used in studies of various influenza vaccines. The use of suitable adjuvants, such as aluminum-based adjuvants, oil-in-water adjuvants such as MF59, adjuvant system 03, (AS03), or toll-like receptor (TLR)-based adjuvants such as those targeting TLR4, or the combination of different adjuvants, could also help to increase antigen immunogenicity and vaccine efficacy [153,172]. Finally, UIVs against antibody-based traditional vaccines will require large-scale preclinical and clinical comparative studies, as well as the standardization of vaccine production and delivery protocols for commercial UIV production.

## Figures and Tables

**Figure 1 viruses-13-00973-f001:**
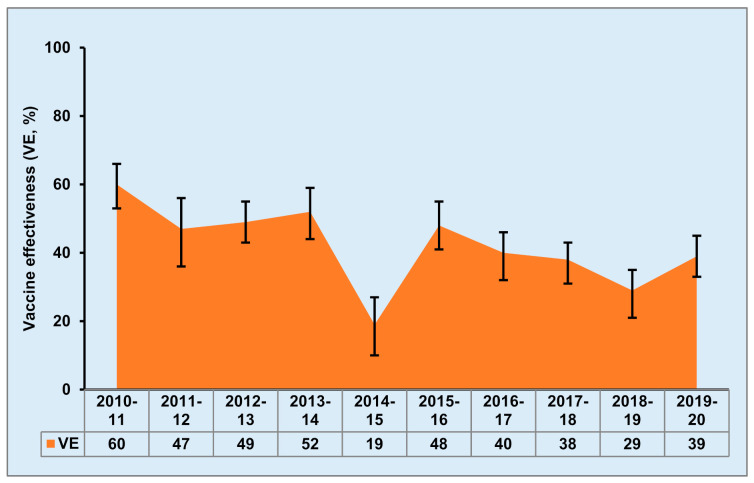
Estimated vaccine effectiveness (VE) for flu seasons from 2010–2020. Data is adapted from the Centers for Disease Control and Prevention seasonal flu vaccine effectiveness studies [10]. Vaccine effectiveness which defines as the percent reduction in the frequency of influenza illness among vaccinated people compared to non-vaccinated people, is estimated using data from the United States vaccine effectiveness Network. Data are presented as adjusted overall VE (%) with 95% confidence intervals.

**Figure 2 viruses-13-00973-f002:**
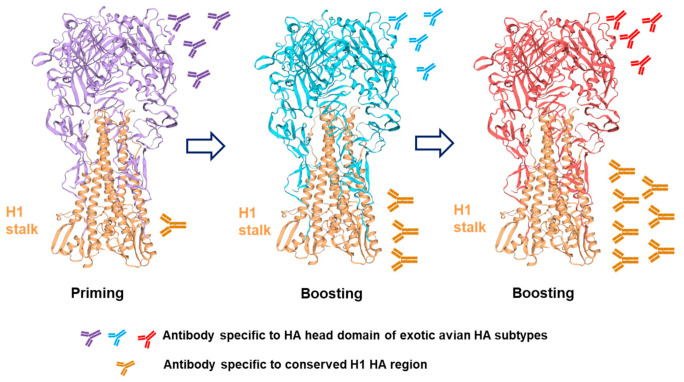
Chimeric HA vaccination approach. In this example, the cHA vaccine contains a stalk domain from H1 HA and head domain from an exotic avian HA subtype. Priming with the cHA induces a low antibody (Ab) response to stalk H1 HA. However, boosting with other cHAs containing the same stalk H1 HA domain with different HA head domains of exotic avian subtypes increases Ab responses to the H1 HA stalk domain. The protection conferred by the cHA is mediated by both neutralizing and non-neutralizing Abs (such as ADCC). The structure of HA was constructed using Swiss-Model (https://www.swissmodel.expasy.org/, accessed on 5 April 2021). The representative H1 HAs are from influenza A virus (A/California/07/2009(H1N1)) with sequence obtained from GenBank (ACQ55359.1) or from influenza A virus (A/goose/Guangdong/1/1996(H5N1)) with sequence obtained from GenBank (YP_529486.1).

**Figure 3 viruses-13-00973-f003:**
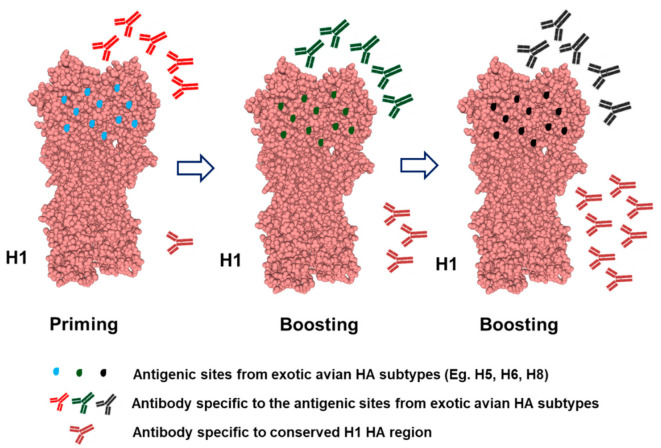
Mosaic HA vaccination approach. The mHA vaccine contains a head domain in which the variable immunodominant antigenic sites are replaced with antigenic sites from exotic avian HA subtypes while conserved regions of both head and stalk domain are retained (H1 HA in this example). Priming with mHA strongly induced an Ab response to the conserved antigenic sites, but only a weak Ab response to the conserved retained regions. However, boosting with the mHA head domain with antigenic sites from other exotic avian HA subtypes enhances Ab responses to both the head and stalk domains. The structure of HA was constructed using Swiss-Model (https://www.swissmodel.expasy.org/, accessed on 5 April 2021). The representative virus is influenza A virus (A/California/07/2009(H1N1)) and the sequence was obtained from GenBank (ACQ55359.1).

**Table 1 viruses-13-00973-t001:** Various universal Influenza vaccines in clinical trials.

**Immunogen**	Vaccine Name /Platform	Identifier	Manufacturer	Phase	Status	Effect	Adjuvant
HA	Chimeric HA-based LAIV combinations	NCT03300050	Icahn School of Medicine at Mount Sinai (US)	1	Completed	Induces high anti-stalk Ab titers and long-lasting immunity	AS03A
M2e	M2e-based VLPs	NCT00819013	Ghent Univ (Belgium) Sanofi Pasteur (US)	1	Completed	Induces anti-M2e Ab	Alum and QS-21
	RedeeFlu M2SR/M2e-deficient	NCT03999554 NCT02822105 NCT03553940 NCT04785794	Flugen, Inc. (US)	1 1 1 1	Completed Completed Completed Ongoing	Reduces symptom scores and virus replication	None
NP and M1	MVA−NP+M1/Viral vector	NCT00942071 NCT00993083 NCT01465035 NCT01818362 NCT02014168 NCT03277456 NCT03300362 NCT03883113 NCT03880474	Vaccitech Ltd.	1 2 1 1 1 1 2 2 2	Completed Completed Completed Completed Terminated Completed Completed Completed Terminated	Reduces influenza symptoms and length of virus shedding, induces T cell responses	None
	ChAdOx1 NP+M1/Adenoviral vector	NCT01623518	Jenner Institute, University of Oxford	1	Completed	Increases T cell response	None
NP	OVX836/Reco-mbinant NP	NCT04192500 NCT03594890	Osivax SAS (France)	2 1	Completed Completed	None yet reported	None
HA, NP, and M1	M-001/ Recombinant protein	NCT01010737 NCT01146119 NCT01419925 NCT02293317 NCT02691130 NCT03058692 NCT03450915	BiondVax Pharmaceuticals Ltd. (Israel)	1/2 2 2 2 2 2 3	Completed Completed Completed Completed Completed Completed Completed	Induces significant cellular-mediated immunity and HI titers	Montanide ISA-51/ Oil –in-water
NP, M1, P1, and P2	FP-01.1/ Peptide based	NCT01265914 NCT01677676 NCT01701752 NCT02071329	Immune Targeting Systems Ltd. (United Kingdom)	1 1 1 1/2	Completed Completed Completed Completed	Good safety and tolerability profiles	None
NP, M1, and M2	FLU-v/ Peptide based	NCT01181336 NCT01226758 NCT03180801 NCT02962908	PepTcell (SEEK, United Kingdom)	1 1 2 2	Completed Completed Completed Completed	Stimulates cell-mediated immunity, reduces symptomatology and virus shedding	Montanide ISA-51/Oil –in-water

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
