# Peer review of "Targeting Antigens for Universal Influenza Vaccine Development"

_viruses, 2021, doi:10.3390/v13060973_

Round 1
Reviewer 1 Report
The manuscript “Targeting antigens for universal influenza vaccine development” by Nguyen is an extensive review with all the available literature reviewed on the development of novel universal vaccines. Authors have covered most areas of current and progressing research and summarized them in a very constructive manner. My comments and suggestions to improve the manuscript is detailed below:
- Line 79: Define VE, how is the effectiveness is calculated?
- Line 157: May add more information on mosaic HAs, such as mosaic” HAs (mHAs) were designed with the idea of eliciting antibodies not only against the conserved stalk domain but also against epitopes in the head domain outside of the major antigenic sites.
- Line 99-100, please add the reference next to the citation of authors, Impagaliazzo and Yassine.
- Line 266-275 may be moved to Line 253, for continuity of the data presented.
- If possible, provide a Table of current vaccines and future vaccines under study, any vaccines on Phase 1,2 or 3 etc.
- After each section, please provide the advantage and disadvantage in couple of sentences and authors suggestions for a future research of that UIV. Something similar to Line 339-347.
- Conclusion, should be modified to reflect the prospect of any UIV with a promising future research or suggestion to improve the UIV research.
Author Response
We would like to thank the editorial office and the reviewers for giving us the chance to revise important areas of our manuscript. We have carefully addressed the concerns that the reviewer raised for the improvement of our paper, and include them in appropriate sections of the manuscript to which these revisions apply. Below we highlighted sections in the text corresponding to our response to the reviewer’s comments. We hope that our revised manuscript provides the format and details acceptable for publication in Viruses.
Thank you very much for your consideration of our manuscript.
Response to Reviewer 1’s comment
Comments and Suggestions for Authors:
The manuscript “Targeting antigens for universal influenza vaccine development” by Nguyen is an extensive review with all the available literature reviewed on the development of novel universal vaccines. Authors have covered most areas of current and progressing recsearch and summarized them in a very constructive manner. My comments and suggestions to improve the manuscript is detailed below:
Point 1. Line 79: Define VE, how is the effectiveness is calculated?
Response 1. We appreciate the Reviewer’s critical comment.
According to the CDC, two general types of studies are used to measure how well an influenza vaccine works, including observational studies and randomized controlled trials. Vaccine effectiveness (VE) refers to vaccine protection measured in observational studies. VE is the percent reduction in the frequency of influenza illness among vaccinated people compared to non-vaccinated people, usually with adjustment for factors that are related to both influenza illness and vaccination.
This information has been added on lines 78-79, page 2 in the revised manuscript.
“Vaccine effectiveness which defines as the percent reduction in the frequency of influenza illness among vaccinated people compared to non-vaccinated people.”
Point 2. Line 157: May add more information on mosaic HAs, such as mosaic” HAs (mHAs) were designed with the idea of eliciting antibodies not only against the conserved stalk domain but also against epitopes in the head domain outside of the major antigenic sites.
Response 2. We appreciate the Reviewer’s insightful comment. We have added the suggested sentence to the line 165-167, page 4 and 5.
“Mosaic HAs (mHAs) were designed with the idea of eliciting antibodies not only against the conserved stalk domain but also against epitopes in the head domain outside of the major antigenic sites.”
Point 3. Line 99-100, please add the reference next to the citation of authors, Impagaliazzo and Yassine.
Response 3. We appreciate the Reviewer’s comment. We have added proper references next to the citation of authors as references 23 and 24 in line 102, page 3.
“In 2015, Impagliazzo et al. [23] and Yassine et al. [24] independently generated stable and correctly folded headless HA by introducing a stabilizing element in the mem-brane-proximal end of the molecule.”
Point 4. Line 266-275 may be moved to Line 253, for continuity of the data presented.
Response 4. We appreciate the Reviewer’s insightful comment. As suggested, we have moved the paragraph to the line 285-293, page 7.
Point 5. If possible, provide a Table of current vaccines and future vaccines under study, any vaccines on Phase 1,2 or 3 etc.
Response 5. We appreciate the Reviewer’s highly constructive comment. We have added table 1 which contains information regarding various universal vaccines in clinical trials on Page 12.
“Table 1. Various universal Influenza vaccines in clinical trials.”
Point 6. After each section, please provide the advantage and disadvantage in couple of sentences and authors suggestions for a future research of that UIV. Something similar to Line 339-347.
Response 6. We appreciate the Reviewer’s constructive comment.
Considering the Reviewer’s comment. we have added a couple of sentences for the advantage and disadvantages of each UIV and our suggestion for future research of UIV as following:
Page 4, lines 147-152:
“Although cHA-based vaccines could induce Abs against anti-stalk HA and provide protection against both homologous and heterologous viral challenges, the protective efficacy of this vaccine might still be limited depending on the chimeric antigen subtypes. Further, multiple-dose vaccinations of different chimeric vaccines are required to induce vaccine efficacy with a broad spectrum. Therefore, planned antigen selection and simplified vaccine strategies are needed for further study.”
Page 6, line 205-209:
“Although mHA-based vaccines could provide broader cross-reactive protection against both homologous and heterologous viral challenges, similar to cHA-based vaccines, the protective efficacy of this vaccine might be still limited depending on multi-dose vaccination and the selected antigen subtypes. Therefore, planned antigen selection and simplified vaccine strategies are needed for further study.”
Page 6, 233-238:
“Overall, the COBRA HA approach is one of the promising strategies to develop UIVs by generating broadly reactive anti-head HA Abs to protect against a broad range of influenza viruses. However, sometimes the generated consensus sequences do not generate fully functional HA proteins, and thus, it needs to use alternative vaccine platforms, such as VLP. Currently, COBRA HA vaccines are in the late preclinical stage of development, and clinical trial studies are needed to investigate their vaccine effects in humans.”
Page 7; line 267-277:
“Collectively, these studies indicate that hyperglycosylation could be a promising strategy to find hidden epitopes in the HA head and stalk domains, facilitating the development of novel UIVs in the future. Currently, glycosylation in influenza vaccine design is still under debate. Studies showed that glycosylation was important for virus infectivity and host immune responses. Wu et al. reported that glycosylation of HA at N-142 promoted H1N1 influenza virus infectivity and the glycosite 142 was important for the virus in evading immune responses in humans [70]. The addition of N-link glycosylation modification to the head HA region also decreases Ab titers in HI assay and plaque-reduction neutralization test [71]. Thus, it is necessary to determine the exact glycosites needed for influenza vaccine development. Various studies should also be performed to have a better understanding of the influence of glycosylation on vaccine efficacy.”
Page 8; line 316-319:
“To our knowledge, there is no clinical trial of NA-based vaccines carried out so far; thus, the development of the vaccine becomes more challenging and requires more effort to make them available. Clinical trials and observational studies of NA-based vaccines should be performed to evaluate their vaccine effects in humans.”
Page 10; line 431-438:
“Although internal protein-based influenza vaccines showed promising results concerning the induction of T cell immune responses and cross-protection, however, internal protein-based vaccines weakly induce Ab response, which is needed to block influenza virus infection. In addition, a major challenge in developing T cell-based vaccines based on internal protein is the diversity of HLA haplotypes responsible for antigen peptide binding and T cell presentation. Thus, to produce effective internal protein-based vaccines that provide sufficient coverage to all individuals with different HLA diversity, there is a need to further evaluate the response in the context of ethnicity.”
Page 11, 478-484:
“Overall, the multiple proteins/peptides-based vaccine approach is one of the promising strategies to develop UIVs by combinations of multiple proteins or peptides, but there are still limitations, such as the sequence variations of target proteins among the subtypes. Therefore, sequences of target proteins in designed vaccine candidates should be screened to provide broad protection against different virus strains. The immunodominant regions should also be well selected to obtain the most appropriate epitopes for vaccine construct.”
Point 7. Conclusion, should be modified to reflect the prospect of any UIV with a promising future research or suggestion to improve the UIV research.
Response 7. We appreciate the Reviewer’s insightful comment. As suggested, we have revised our conclusion with the prospects of each UIV and suggestions to improve the UIV research on Page 13, lines 533-547.
“Although NA-based vaccines have shown induction in immune responses, NA is currently ignored as an antigen target in influenza vaccine development due to its limitations, such as standardization issues. Further studies of NA-based influenza vaccines should be performed to demonstrate the important roles of NA as an influenza vaccine antigen. Also, advanced technologies should be developed to quantify the absolute amount of NA and measure NA Ab responses accurately and maintain neuraminidase activity of the NA-based vaccines during storage. Although M2e, a well-conserved protein of influenza viruses, is a promising target for UIVs, the M2e protein has low immunogenicity. Therefore, to induce robust immune responses, it should be used with other viral proteins or adjuvants. In addition, the internal protein-based vaccines, that mainly induce T cell responses, facilitating the killing of influenza-infected cells, is one of the promising strategies to develop UIVs by combinations of multiple proteins or peptides; however, these vaccines weakly induce Ab response, hereof the immunodominant regions should be properly selected to obtain the most appropriate epitopes for vaccine construct.”
Reviewer 2 Report
This manuscript reviews the literature concerning the scientific approaches to the development of universal influenza vaccines (UIV). It focuses on the types of antigens used. Overall, it has summarized clearly the current state of the science in the field. Although it has not been able to offer conclusions on the promised approaches, it has done a good job of describing the overall picture.
My suggestion is that the modifications of glycosylation of these antigens offer an intellectually exciting approach. However, the current states appears to be contradictory among the various studies, with some studies suggesting the opposite roles of some types of glycosylation. Perhaps it will be worthwhile to devote more space in this discussion.
Author Response
We would like to thank the editorial office and the reviewers for giving us the chance to revise important areas of our manuscript. We have carefully addressed the concerns that the reviewer raised for the improvement of our paper, and include them in appropriate sections of the manuscript to which these revisions apply. Below we highlighted sections in the text corresponding to our response to the reviewer’s comments. We hope that our revised manuscript provides the format and details acceptable for publication in Viruses.
Thank you very much for your consideration of our manuscript.
Response to Reviewer 2’s comment
Comments and Suggestions for Authors:
This manuscript reviews the literature concerning the scientific approaches to the development of universal influenza vaccines (UIV). It focuses on the types of antigens used. Overall, it has summarized clearly the current state of the science in the field. Although it has not been able to offer conclusions on the promised approaches, it has done a good job of describing the overall picture.
My suggestion is that the modifications of glycosylation of these antigens offer an intellectually exciting approach. However, the current states appears to be contradictory among the various studies, with some studies suggesting the opposite roles of some types of glycosylation. Perhaps it will be worthwhile to devote more space in this discussion.
Response: We appreciate the Reviewer’s insightful comment. As suggested, we have added some recent debates on vaccine efficacy of modified glycosylation sites on HA protein on page 7, lines 267-277.
“Currently, glycosylation in influenza vaccine design is still on debate. Studies showed that glycosylation was important for virus infectivity and host immune responses. Wu et al. reported that glycosylation of HA at N-142 promoted H1N1 influenza virus infectivity and the glycosite 142 was important for the virus in evading immune responses in humans [70]. The addition of N-link glycosylation modification to the head HA region also decreased Ab titers in HI assay and plaque-reduction neutralization test [71]. Thus, it is necessary to determine the exact glycosites needed for influenza vaccine development. Various studies should also be performed to have a better understanding of the influence of glycosylation on vaccine efficacy”